# Imaginary Coating Algorithm Approaching Dense Accumulation of Granular Material in Simulations with Discrete Element Method

Fei Wang [1], Yrjö Jun Huang[2,3,*] and Chen Xuan [4]

1 College of Physics and Electromechanics Engineering, Hexi University, Zhangye 734099, China
2 Department of Aeronautics and Astronautics, Fudan University, 220 Handan Road, Shanghai 200437, China
3 Suzhou Jiyi BioTechnology Inc., Suzhou 215000, China
4 Department of Foundational Mathematics, Xi'an Jiaotong-Liverpool University, Suzhou 215000, China
* Correspondence: jun_huang@fudan.edu.cn; Tel.: +86-156-1869-2690

**Abstract:** The difficulty of obtaining a densely packed granular material as an initial condition is a very common problem in numerical simulations of granular materials. In this article, an Imaginary Coating Algorithm (ICA) is introduced. To avoid unreasonable particle deformation when using a longer time step and a lower Young's modulus, the radii used in calculating the action forces in a binary collision are slightly larger than the real values. In other words, an imaginary coat is added to each particle or element. To validate this algorithm, simulations were carried out by using ÅDEM, and a A Discrete Element Method (DEM) software program was developed. Compared with traditional Simulated Annealing Algorithms (SAA), this technique can approach the densely packed state with less CPU/GPU time and is easy to operate.

**Keywords:** Imaginary Coating Algorithm; Discrete Element Method; granular materials; simulated annealing method

## 1. Introduction

Granular packed beds are widely used in chemical engineering fields. To obtain a description of the packed bed structure and the void fraction profile, a large amount of work was carried out [1–3]. A relatively recent method to model a particulate system is the Discrete Element Method (DEM). Generally speaking, due to the relative independence of particles in mesh-free algorithms, such as the Material Point Method (MPM), Smoothed Particle Hydrodynamics (SPH), and DEM, these mesh-free algorithms are more suitable for GPU computing. However, when dealing with the dense accumulation problems due to gravitation using the DEM, the literature is scarce [3]. One possible reason is that the DEM is CPU/GPU time-consuming.

Recarey et al. summarized their methodology for establishing a particle densely packing state [4]. According to their introduction, a Monte-Carlo method is used to form a nucleus with a few particles. Then, new particles are allocated tangentially to the growing nucleus on this nucleus, until the entire space is filled. In contrast, natural accumulation using gravitation may be a simpler and cruder method, especially for 3-D problems. The main process of their method is given as follows. The location of each particle is created by the Monte Carlo method, and no particle in the initial state is overlapped with others. An acceleration field is added to the system to obtain a packing state. To accelerate convergence in DEM simulations, a Simulated Annealing Algorithm (SAA) is employed in Pickett et al.'s work [5]. Another SAA is introduced by Huang [6], in which an exaggerated acceleration is added to the system. After the system approaches a densely packed state, the acceleration decreases to the appropriate level. Accompanying this exaggerated force field added, shaking and vibration are also simulated to obtain a higher packing density [7]. Although

both of these algorithms can lead to a solution with high accuracy, they are still CPU time-consuming. To obtain a rough solution in a shorter time, a new algorithm, the Imaginary Coating Algorithm (ICA), is introduced in this article. Of course, the SAA can also be used based on the results from the ICA, which may lead to a denser accumulation with particle deformation than when using the ICA only.

The structure of this article is as follows: The Glued Particle Method (GPM) is introduced in Section 2, which is used to approach complex-shaped particles in this article. Then, a commonly used constant coefficient of restitution (COR) in the contact model for DEM simulation is introduced in Section 3. The new algorithm developed in this article is introduced in Section 4. Based on the content presented in Sections 2–5, a numerical simulation is presented, and the results are compared with those from the traditional SAA. The last section presents the conclusions.

## 2. Glued Particle Method

Several methods have been developed to describe complex-shaped, (non-circular/spherical) particles, such as the GPM [8], polyhedra method [9] and super-ellipsoid method [10,11]. In this article, we employ the GPM to connect several overlapped circular elements for one elliptical particle [12].

As illustrated in Figure 1, the circular element, $O_i$, is called a host element, on which the mass center of the particle is located, while the other two elements, $O_i'$ and $O_i''$, are called slave elements. In a collision between elliptical particle(s), the forces acting on one elliptical particle are as plotted in Figure 1a. In this figure, the other particle can be either a circular particle or part of a particle of a complex shape. Here, only the motion of and forces acting on the slave elements are discussed. The total force, $\mathbf{F}$, should be decomposed into two independent components that are orthogonal to each other, when calculating the motion of the elliptical particle. The normal force is along the direction from the contact point $c$ to the mass center of the elliptical particle, $O_i$, as shown in Figure 1b, with the magnitude of normal force $|\mathbf{F}_n'| = |\mathbf{F}_n| \sin(\beta + \chi)$.

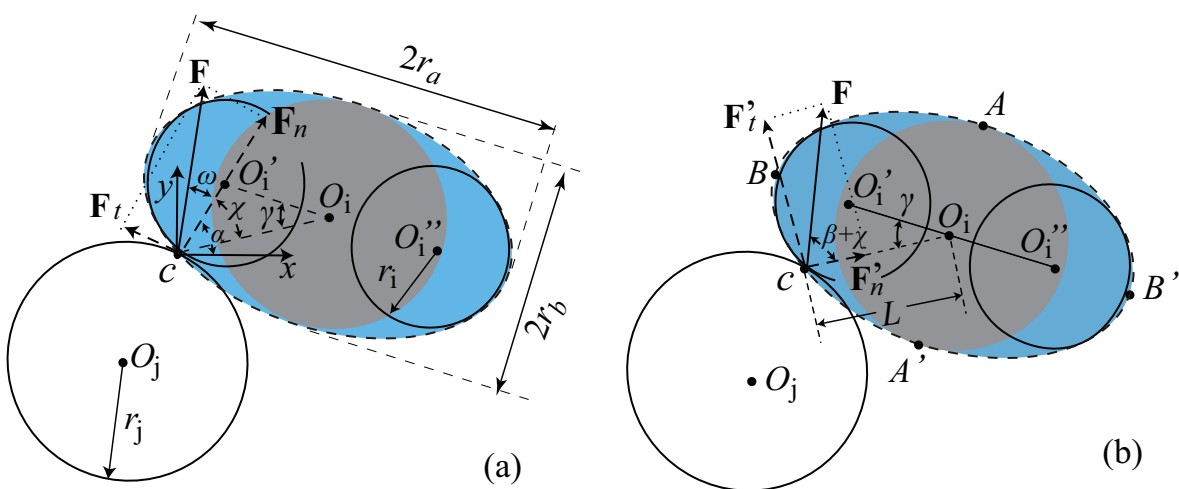

**Figure 1.** Sketch of a collision with one elliptical particle. The elliptical particle is composed of three circular elements and $c$ is the contact point. (**a**) Normal force $|\mathbf{F}_n|$ and tangential force $|\mathbf{F}_t|$ are obtained from the binary collision of circular particles $O_i$ and $O_j$; (**b**) the total force, $\mathbf{F} = \mathbf{F}_n' + \mathbf{F}_t'$, is decomposed into $\mathbf{F}_n'$ and $\mathbf{F}_t'$ to calculate the motion of the elliptical particle $O_i$.

There are several time integration algorithms to be chosen, such as the Standard Euler Method and the Verlet Method [13]. Among these algorithms, the Standard Euler Method is one of the simplest algorithms with first-order precision. The linear velocity and the angular velocity are given by

$$\mathbf{u}' = \mathbf{u} + \frac{\sum \mathbf{F}}{m} \tag{1}$$

$$\mathbf{\Omega}'_{n,t} = \mathbf{\Omega}_{n,t} + \frac{\sum (\mathbf{F} \times L)}{I} \tag{2}$$

respectively, where the primes are used for parameters at the end of a time step, while those without primes are used for the beginning of the time step. $I$ is the moment of inertia. For ellipses, $I = (r_a^2 + r_b^2)/5$ and $\mathbf{L}$ is the vector from the mass center to the contact point, $c$, i.e.,

$$|L| = [(x_c - x_{O_1})^2 + (y_c - y_{O_1})^2]^{\frac{1}{2}} \tag{3}$$

The details of the GPM used by us can be found in Refs. [7,14].

## 3. Brief Introduction to Collision Model

The DEM was first introduced by Cundall for the analysis of rock mechanics [15], and it was extended to soil mechanics by Cundall and Strack [16] in the 1970s. Furthermore, the DEM can be divided into the Event-Driven Method (EDM) and the Time-Driven Method (TDM). The Event-Driven Method is also called the hard particle model. There is an event, e.g., a collision inside the system, which controls the system dynamics. Coefficients of restitution are used to describe the differences before and after the event, while the driving force in the TDM is the contact force among particles. The TDM allows finite deformation between two discrete particles in contact and is able to detect contacts among particles automatically. Hence, the TDM is also called the soft particle model. Generally speaking, the EDM is suited for dilute flow, but with a longer time integration step, while the TDM can be used for dense particulate flow systems with a shorter time integration step. Moreover, in order to describe the interaction forces among particles, there are many models to be chosen, such as the Johnson–Kendall–Roberts (JKR) model [17], the Derjaguin–Muller–Toporov (DMT) model [18], and the van der Waals model [19].

In this article, a particle–particle interaction model given by Ramirez et al. [20] is used. In this model, a contact process could be modeled as a spring–dashpot system, as shown in Figure 2. Here, the role of the dashpot is to ensure that the kinetic energy of the system is dissipated after particle–particle collisions. In this model, both motion in the normal and the tangential directions is included, which is represented by the subsripts $n$ and $t$, respectively. In this article, only normal motion is discussed. Of course, this can be easily extended to problems with tangential components. This normal force, introduced by Ramirez et al. [20], is the sum of an elastic term and a viscous term at the contact point, namely

$$\begin{aligned} F_n &= F_n^e + F_n^d \\ &= K_n \delta_n^\zeta + \eta_n \delta_n^\xi \dot{\delta}_n \end{aligned} \tag{4}$$

where $\delta_n$ is the overlap, $\delta_n = r_i + r_j - l_{ij}$ $(\delta_n > 0)$, and the subscripts $i$ and $j$ indicate the two colliding particles, respectively. Otherwise, $F_n = 0$ when $\delta_n < 0$. Here, $r_{i,j}$ is the radius of each circular element composing an elliptical particle and $l_{ij}$ is the distance between two centers of the circular particles. $K_n$ is the effective stiffness coefficient and $\eta_n$ is the damping coefficient. By means of dimensionless analysis of the Hertz theory, it is shown that $\zeta = \frac{3}{2}$ and $K_n = \frac{4}{3} Y_r \sqrt{r_r}$ [21]. In addition, Kuwabara and Kono pointed out that $\xi = 0.5$ [22]. For a constant coefficient of restitution (COR) in the normal direction, $e$, the damping coefficient $\eta_n$ is given as a function related to the constant COR in the normal direction, $e$, i.e., $\eta_n = -2\kappa \sqrt{\frac{5k_n m_r}{6}}$ [23]. Here, $Y_r$ is the effective Young's modulus given by $Y_r = \frac{Y}{2(1 - v^2)}$, in which $Y$ is the Young's modulus, $v$ is the Poisson's ratio, and $r_r$ and $m_r$ are the effective radius and effective mass, defined as $r_r = \left( \frac{1}{r_i} + \frac{1}{r_j} \right)^{-1}$ and $m_r = \left( \frac{1}{m_i} + \frac{1}{m_j} \right)^{-1}$, respec-

tively. In addition, $\kappa$ is the damping coefficient, i.e., $\kappa = \dfrac{\ln(e)}{\sqrt{\pi^2 + \ln^2 e}}$, and $k_n$ is given by $k_n = 2Y_r \sqrt{r_r}$. The damping coefficient $\eta_n$ is a function of masses $m_{i,j}$, $e$, $r_r$ and $Y_r$ [23]. For an oblique collision, there are two commonly used methods to obtain the magnitude of the tangential force $\mid \mathbf{F}_{t,i} \mid$. One is obtained from Coulomb's law, i.e.,

$$\mid \mathbf{F}_{t,i} \mid = \mu \mid \mathbf{F}_{n,i} \mid, \tag{5}$$

where $\mu$ is the coefficient of friction, which is a constant in this model. The other method is to use the spring–dashpot model in the tangential direction, as shown in Figure 2. The equation for the tangential direction is similar to that for the normal direction—namely, Equation (5) [24].

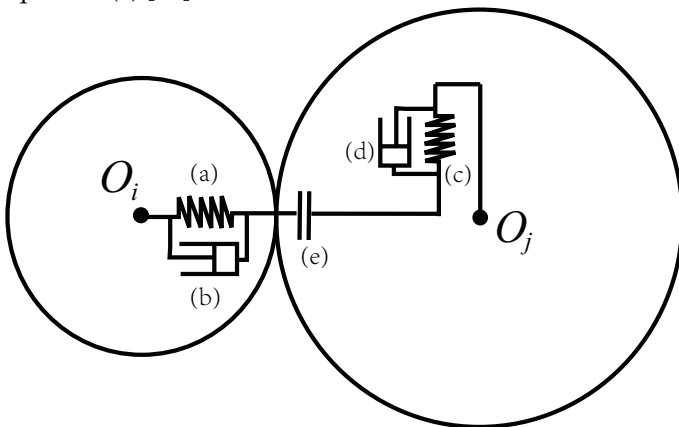

**Figure 2.** A viscoelastic binary collision model includes (**a**) a spring for the normal direction; (**b**) a dashpot for the normal direction; (**c**) a spring for the tangential direction; (**d**) a dashpot for the tangential direction; and (**e**) a contact judgment switch.

Here, it should be emphasized that the constant COR can only be used for spherical/circular particles. In the case of particles with a complex profile, some of the rebound kinetic energy may be converted to rotational kinetic energy. If a constant COR is adopted, the energy may be divergent after a collision. Huang et al. gave some preliminary results on the modeling of the kinetic energy allocation for collisions between ellipsoidal particles [25,26]. In this article, a constant COR is still employed, but the COR should be small enough to ensure the convergence of energy.

## 4. Imaginary Coating Algorithm

To speed up a DEM simulation of the accumulation process due to gravitation using the collision model introduced previously, a lower $Y$ could be used, resulting in a longer collision time, $t_c$, and a longer integral time step, $dt$, where $t_c \propto Y_r^{-0.4}$ [21] and $dt \propto t_c$ [27]. Following Shen and Sankaran's suggestion [28], the collision time is given as $t_c = \dfrac{\pi}{\sqrt{(2K_n/m_r)(1 - \kappa^2)}}$, where $K_n$ is a function of the Young's modulus and the radius of the smallest particle in the system, $r_{min}$. In addition, to avoid any unreasonable overlap, $dt$ should also be smaller than the Rayleigh time, $dt_R$, i.e., $dt_R = \dfrac{\pi r_{min}}{\gamma} \sqrt{\rho/G}$, where $\rho$ is the density, $G$ is the shear modulus, $\sqrt{\rho/G}$ is the the propagating velocity of a Rayleigh wave, $\gamma = 0.8677 + 0.163v$, and $r_{min}$ [29].

The lower $Y$, the softer the particle is, and a greater overlap happens in a collision. In other words, to converge the velocities of the particles to zero as soon as possible, softer particles are employed initially in the simulation and these particles return to their normal hardness at a reasonable rate in Pickett et al.'s algorithm [5]. For the steady state, the value of COR does not affect that for the overlap. On the other hand, Huang's algorithm [6] increases the forces and leads to greater overlaps. Thus, all gravitational potential energy is transformed into deformational potential energy. However, the disadvantage of these algorithms is that the changed COR or force field cannot be recovered to

the actual values immediately. Otherwise, the released deformational potential energy is huge, and all particle clusters are scattered.

To overcome this shortage, the radius used in the simulation, $r + d_c$, is slightly larger than that of the real particle, $r$, as shown in Figure 3. For a viscoelastic collision, the viscous term becomes zero when particles are densely packed, if there is no relative velocity between these particles in contact. From Equation (1), from the relationship between the external force $F_n$ and the overlap $\delta_n$, it can be obtained that

$$\delta_n = \left(\frac{3F_n}{4Y_r}\right)^{\frac{2}{3}} \left(\frac{r}{2}\right)^{\frac{1}{3}}. \tag{6}$$

On the other hand, the particle interaction force varies with depth, $D$. The normal force between two neighboring particles for a 1-D spherical particle column of the same size, material and arrangement in the direction of gravitation is given by

$$|F_n| = C\pi r^2 \rho D g, \tag{7}$$

where $C = 2/3$ for a 1-D vertical column, $\rho$ is the density, $g$ is gravitation, and $D$ is the depth of the particle from the top. It can be seen that a significant change in $K$ has little effect on $\delta_n$, or the void fraction. For example, for glass particles with a diameter of 1 cm, the density is 2600 kg/m$^3$ and the Young's modulus $Y = 6.5 \times 10^{10}$ Pa; the error of the void fraction $\varepsilon \ll (\delta^*)^3 \sim 10^{-9}$. Figure 4 depicts the relationship between the dimensionless overlap, $\delta^*$, and $D$, where $\delta^* = \delta/r$. For a packing problem with a characteristic length of 1 m, the void fraction error is less than $1 \times 10^{-12}$ [30]. Here, the abscissa is the depth. Because glass particles with a diameter of $d = 1$ cm are used for the calculation, the depth can be converted into the number of balls, $N$, i.e., $N = h/d$, where $h$ is the depth. Once those particles within the ICA achieve a steady packing state, the real radii $r$ and Young's modulus can be used. With the change in size and Young's modulus, the particle system becomes unstable again. Since the void fraction is very close to the real value and there is not too much extra potential energy, the mechanical energy of the whole system is very small. In other words, particles cannot achieve high velocities if they fall freely without an imaginary coat. In addition, because of the large coordinate number of each particle, the mechanical energy dissipates quickly and approaches a new balanced state, namely a steady packed state. The coordinate number is defined as the number of neighboring particles in contact with the same particle at a given moment.

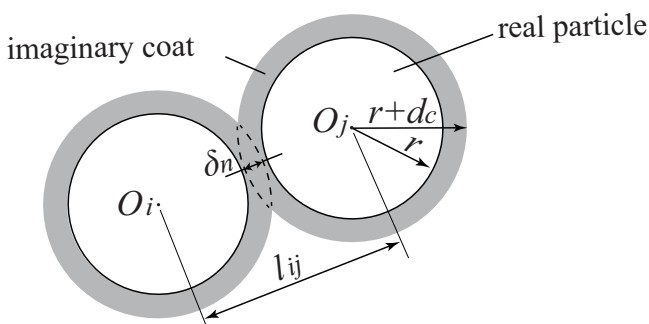

**Figure 3.** Sketch of the ICA. Solid circles show two same-sized real particles and gray parts are the imaginary coats, where $d_c > \delta_n$. The thickness of the imaginary coats, $d_c$, is much greater than the maximum overlap, $\delta_n$, in the system.

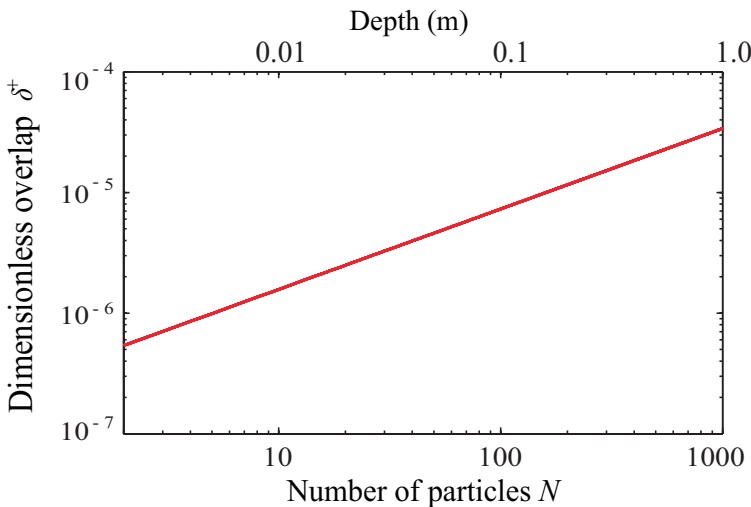

**Figure 4.** The changes in overlap with depth for densely packed glass particles with radii of 0.5 cm. It can be seen that the logarithm of depth, number of particles or external force is proportional to the deformation $\delta_n$.

## 5. Comparison between ICA and SAA

To test the validity of the ICA, 2-D simulations employing both the ICA and SAA introduced by Huang [6] were carried out based on their in-house-developed DEM software, ÅDEM (Open DEM in Norwegian), which is a program based on CPU calculation. In addition, both of the simulations presented in this article are single thread calculations. A total of 3000 elliptical glass particles with semi-major axis $r_a = 7.5$ mm and semi-minor axis $r_b = 2.5$ mm are packed in a container. To shorten the CPU time, a low COR, $e$, is set for all particles, i.e., $e = 0.2$, and the Young's modulus $Y = 6.5 \times 10^{10}$ Pa. If a smaller Young's modulus is used, the collision time increases. For example, $Y = 6.5 \times 10^5$ Pa leads to the collision time being 158.5 times larger than that of $Y = 6.5 \times 10^{10}$ Pa, while the overlap is only approximately 6.3 times that using $Y = 6.5 \times 10^{10}$ Pa. In this packing problem with a characteristic depth of 0.1 m, the error of the void fraction $\varepsilon \ll (\delta^*)^3 \sim 10^{-12}$, where the dimensionless overlapping, $\delta^*$, is defined by $\delta^* = \delta/r$.

The simulation process using the ICA is given in Figure 5, in which Figure 5a shows the initial state of the calculation. Here, a constant downward gravity field is loaded throughout the whole simulation process. All elliptical particles are regularly arranged in the calculation domain. It should be noted that only the lower part of the container is shown in Figure 5a. The geometric profile of each elliptical GPM is approximated using nine circular elements, as shown in Figure 5b. Here, the real radii of those elements composing the elliptical particle are plotted in gray, while the imaginary coats added are shown by the red dashed circles. Of course, this is merely a schematic diagram. In order to converge as soon as possible and reach a densely packed state, in the ICA stage, the COR, $e$, is set to 0.02, and the thickness of the imaginary coats is three times the maximum theoretical thickness, which is a function of the depth, as presented in Equations (6) and (7) and Figure 4. It can be seen that the overlap is not only a function of the depth, but also of the particle size and material properties. Thus, the thickness of the imaginary coat on each element is also related to its radius. In other words, the elements composing the same elliptical particle have imaginary coats of different thicknesses. Thus, there will be no overlapping among particles when the imaginary coats are deleted. In this calculation, Coulomb's law, namely Equation (5), is used to describe the tangential force in collision. Figure 5c is the final calculation result, while a corner of Figure 5c is enlarged in Figure 5d. It can be seen that the ICA has little effect on the accumulation result.

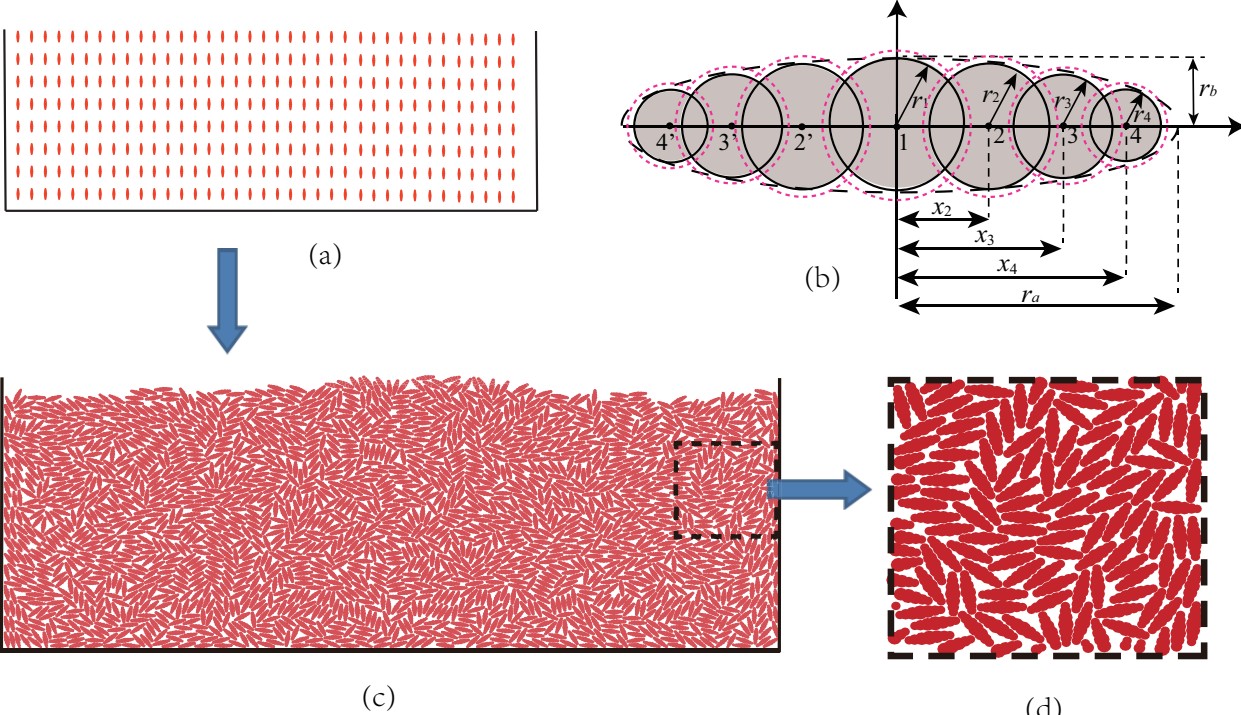

**Figure 5.** Result of the dense accumulation problem of 3000 elliptical particles using the ICA. (**a**) Initial state; (**b**) sketch of the Glued Particle Method (GPM) for an elliptical particle, where the gray profile is the real radius and the red dashed profiles are the imaginary coats; (**c**) final result of dense accumulation; and (**d**) some details of sub-figure (**c**).

The comparison of the results between these two algorithms is plotted in Figure 6. Figure 6a shows the void fraction in the same region given in Figure 5d. It is a square area with an edge length of 100 mm. The void faction, $\alpha$, is calculated by $\alpha = \sum_{i=1}^{N} A_{e,i} / A_s$, where $N$ is an integer indicating the total number of particles in this square area, $A_{e,i}$ is the area of each elliptical particle and $A_s$ is the area of this square. If the center of a particle is located within this square region, the entire area of the particle is considered in this expression of the void fraction. This square area should be far away from the free surface of the accumulated particles, and it also should be large enough to avoid $\alpha > 1$. For the simulation using the SAA (blue curves in Figure 6), a large acceleration of gravitation is given to particles at the beginning and their acceleration values are significantly greater than those using the ICA algorithm (red curves in Figure 6). Thus, the accumulation of the SAA starts earlier than that of the ICA algorithm. However, due to the low COR adopted by the ICA, the energy dissipation of the ICA is faster than that of the SAA when particles are densely packed. Figure 6b shows the kinetic energy of the system. In the process of accumulation, the mechanical energy is rapidly dissipated due to collisions. In order to show the change in kinetic energy clearly, not only are logarithmic coordinates used, but also the total kinetic energy in joules and the dimensionless kinetic energy $E^*$ are plotted in Figure 6b. Here, the dimensionless kinetic energy, $E^*$, is defined as the ratio between the kinetic energy of the whole system at time $t$ and that of the initial state, namely $t = 0$. It should be noted that the void fraction of the simulation using the ICA reaches a stable value around $t = 20$ s. After that, the imaginary coats are removed, and the particle system incurs a small disturbance, which causes a slight decrease in the void fraction. The process without imaginary coats can be considered as a special SAA algorithm, with real gravitation.

The annealing rate is the key problem in simulations employing the SAA. A fast annealing rate may lead to instability and the solution is no longer convergent. On the other hand, a slow annealing rate means a longer CPU/GPU time. In this simulation using

the SAA algorithm, the initial acceleration is 10 times the gravitational acceleration. In the next 36 s, it decreases to the level of one gravitational acceleration at a uniform annealing rate. One of the challenges with the SAA is determining a suitable annealing rate. In the comparison shown in Figure 6, ICA can save more than 30–60% of the CPU time for the same dense accumulation problem shown in Figures 5 and 6 and avoid the time-consuming annealing process, as well as choosing a suitable annealing rate.

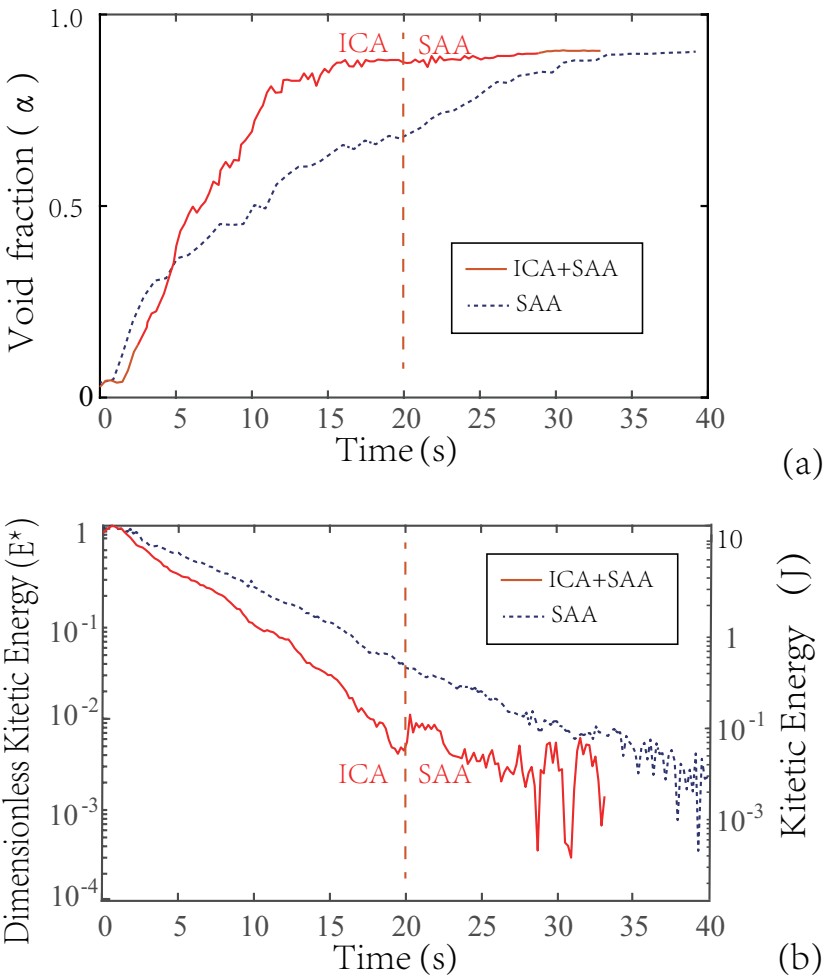

**Figure 6.** A comparison between simulations using the ICA and SAA algorithms. The simulation with the ICA involves two stages. It is simulated with imaginary coats and a low COR, $e = 0.02$, for all particles before $t = 20$ s in the first stage, and without imaginary coats and a normal COR, $e = 0.2$, after $t = 20$ s in the second stage. (**a**) Void fraction and (**b**) kinetic energy of the whole system.

## 6. Discussion

In this article, a new computational algorithm, ICA, to speed up the calculation time for accumulation programs using the DEM is proposed. It is compared with another traditional algorithm, the SAA. Such a comparison is incomplete and general. Although both of these simulations are improved with the same code, as well as the same computer configuration, some of the parameters, such as the thickness of the imaginary coat and the rate of the thickness reduction in the ICA, the reduction rate of gravity in the SAA, namely the annealing rate, and so on, may affect the calculation efficiency. Based on this discussion, the ICA is a new algorithm rather than an optimal solution.

In contrast, the SAA can be considered as increasing the frequency of collisions through virtual potential energy, which increases the energy dissipation rate. On the other hand, the ICA increases the energy dissipation rate through the change in viscosity. Therefore, it is possible to use these two methods simultaneously in the future.

Finally, there is no demonstration of GPU or multi-threaded computation in this article. Both of the simulations in the comparison given in this article are single-threaded. However, since single-threaded computation is the basis of GPU computing and parallel computation, the inference that the ICA is also suitable for GPU or multi-threaded computation is reasonable.

In detail, a DEM simulation has two parts that require CPU or GPU time. One is the neighbor search. In order to speed up this part, background research cells are usually employed. The other time-consuming part is the force/motion calculation between contacting particles pairs. The optimal size of the background grid is related to the velocity distribution and spatial distribution of all particles. In other words, the background research cell size has a great effect on the performance of the simulation with different numbers of CPU/GPU cores or parallel architectures. Therefore, the computational acceleration effect of the ICA depends on the specific situation. It is difficult to give the acceleration ratio quantitatively, but the acceleration effect of the ICA is inevitable.

## 7. Conclusions

The Discrete Element Method (DEM) is widely used in simulations of granular materials. An Imaginary Coating Algorithm (ICA) based on the DEM is introduced in this article. In short, the kinetic energy dissipation rate is accelerated by appropriately enlarging the particle radius, namely the imaginary coat, and reducing its coefficient of restitution (COR). When the kinetic energy is low enough, the original radius and COR are employed, and finally the system achieves a densely packed state with the original Young's modulus. Compared with the Simulated Annealing Algorithm (SAA), this algorithm can not only save CPU/GPU time, but also avoid the troublesome selection of the annealing rate, especially for dense accumulation problems due to gravitation. If the requirement for accuracy is not so high, the results from the ICA are enough, which means that the final result uses large particles with an imaginary coat. Otherwise, the result from this algorithm should be simulated continuously using a decreasing radius, until it reaches the real radius. To validate this algorithm, two simulations of the same accumulation process are presented in this article; one is performed using the ICA and the other is performed using the SAA. The comparison between the ICA and the SAA shows that a higher convergence rate can be obtained by using the ICA.

**Author Contributions:** The idea was devised by Y.J.H.; F.W. performed the simulation and analyzed the results based on Huang's custom-made DEM program ÅDEM. C.X. performed the final data check and revised the language. All authors have read and agreed to the published version of the manuscript.

**Funding:** This research was funded by the National Natural Science Foundation of China (12273004 and 12002287), Jiangsu Sci. & Techn. Program (BK20200248), Jiangsu Univ. Nat. Sci. Res. Program (20KJB130001), Key Program Spec. Fund KSF-E-53 of Xi'an Jiaotong-Liverpool Univ., and the Research Start-Up Foundation from Hexi Univ.

**Institutional Review Board Statement:** Not applicable. This study did not involve humans or animals.

**Informed Consent Statement:** Not applicable.

**Data Availability Statement:** The data presented in this study are available on request from the corresponding author, and the codes of ÅDEM can be found at https://github.com/mmjhuang/, accessed on 1 March 2023.

**Conflicts of Interest:** The authors declare no conflict of interest.

## Abbreviations

The following abbreviations are used in this article:

| | |
|---|---|
| COR | Coefficient of Restitution |
| DEM | Discrete Element Method |
| DMT | Derjaguin–Muller–Toporov Model |
| EDM | Event-Driven Model |
| GPM | Glued Particle Method |
| ICA | Imaginary Coating Algorithm |
| JKR | Johnson–Kendall–Roberts model |
| MPM | Material Point Method |
| SAA | Simulated Annealing Algorithm |
| SPH | Smoothed Particle Hydrodynamics |
| TDM | Time-Driven Model |

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
