# Peer review of "Imaginary Coating Algorithm Approaching Dense Accumulation of Granular Material in Simulations with Discrete Element Method"

_2674-0516, doi:10.3390/powders2010014_

Round 1

Reviewer 1 Report (Previous Reviewer 3)

Lines 134,135: Revise the following statement: "In addition, it is required that to avoid any unreasonable overlap, where rmin is the radius of the smallest particle in the system."

"rmin" is not used in the formula shown in Line 132.

See my comment to the original manuscript (Lines 90-93: Confirm the text. It seems that the formula is missing.).

Author Response

Dear review,

   Thanks for your comments. According to your comments, we revised our draft and marked in blue.  The detail of our point-by-point repsonse  are given as below:

1.   Lines 134,135: Revise the following statement: "In addition, it is required that to avoid any unreasonable overlap, where rmin is the radius of the smallest particle in the system." "rmin" is not used in the formula shown in Line 132.

We checked the context. There is indeed a problem with the expression of the previous version. In this new revised version, we have changed the expression, that Kn is a function of Young's modulus Y and  r_{min}.

2. See my comment to the original manuscript (Lines 90-93: Confirm the text. It seems that the formula is missing.).

we did not find any problem near lines 90-93.  We guess  an equation is expected to appear after "... model given by Ramirez et al. [20]".  Ramirez et al. 's expression is given in eq.4 and therefore we highlingted the context and cite ref.[20] again in blue near line 100. 

Reviewer 2 Report (New Reviewer)

The authors presented an interesting topic worth researching. I think that the ICA code or the calculation algorithm should be made available as a supplement to the article in order to fully understand the details. Unfortunately, the text requires significant language corrections and intensive checking by a native speaker. Attached is the manuscript with the errors I've spotted highlighted.    

Author Response

Dear viewer,

    Thanks for your comments. We revised our draft following your pdf file.  Those modifications according to your requirements are marked in brown. In addition to your comments, we found a native speaker of English to help us improve our language. All these changes are marked in red.

Best Regards

Y.J. Huang 

Round 2

Reviewer 2 Report (New Reviewer)

In my opinion the revised version of the manuscript can be published

This manuscript is a resubmission of an earlier submission. The following is a list of the peer review reports and author responses from that submission.

Round 1

Reviewer 1 Report

In this article, the computational performances of the multi-sphere method (glued particle) and the proposed imaginary coating algorithm are compared in a simulation of particle packing. Although the proposed method may be useful in the context of granular systems modelling, this paper does not provide adequate analyses of the proposed method. For instance:

1. Only one application case (packing of the bulk of particles) is presented and discussed. Even for this application case, we do not see any figures on the temporal dissipation of the kinetic energy of the models.

2. The model is not verified or validated. For example, the model should be tested on a single particle having a collision with a wall at different coefficients of restitution and the dissipation of kinetic energy should be measured by reporting the post-impact / pre-impact velocity. 

3. The models should be compared in other application cases. For example, in hopper flow and silo discharge. Can the proposed model predict the formation of the angle of repose on the surface of particles during silo discharge? Another interesting test case could be the free surface angle in a rotating drum.

4. No strong or weak scaling analyses are provided. The readers do not understand how the computational time changes versus the number of particles, and the computational cost of the model in parallel simulations on several computational cores.

My minor comments:

1. In equation 1, the dissipative force is generally presented as a function of relative velocity between the particles. Why is the dissipative force a function of normal overlap here?

2. Page 3, "To speed a DEM simulation using the collision model introduce ...", This assumption is only valid in gravity-driven flows. In other applications, it may cause non-negligible errors.

3. I do not understand Figure 3. The autors should elaborate more on explaining this figure. Why do we see two x axes for this figure? How does the number of particles correspond to the depth? 

Author Response

Response to Reviewer 1 Comments

   We appreciate the comments from all reviews. We improved our manuscript according to these comments of each reviewer and marked them with different colors. Among these changes, those related to your comments are marked in red. We hope that through these changes, the quality of the article can meet your requirements.

1. Only one application case (packing of the bulk of particles) is presented and discussed. Even for this application case, we do not see any figures on the temporal dissipation of the kinetic energy of the models.

Figure 6(b) is added to show the change of kinetic energy. In addition, the last paragraph of page 7 (lines 225-235) is also added to describe the new figure. Of course, the caption of Figure is also polished.  Corresponding to this new added figure, the description of Figure 6 in line 213,page 7, was also rewritten.

2. The model is not verified or validated. For example, the model should be tested on a single particle having a collision with a wall at different coefficients of restitution and the dissipation of kinetic energy should be measured by reporting the post-impact / pre-impact velocity. 

The assumption of constant coefficient of restitution is questionable. Many literatures have reported that CoR is related to particle size, impact velocity and so on. For complex shaped particle, both particle shape and rotation may affect COR too. We have several articles to explain our new model of non-spherical particles, e.g. ref.[26], which is newly added. Because these models have little relationship with the theme of this article. Therefore, constants COR are used here, or it is averaged based on statistical significance. Since the COR is related to the contact position and relative velocities between particles, only small COR can be used to ensure the convergence of kinetic energy.  A very short explanation to reply this comment is added in Lines122-127, near the bottom of page 4.

3. The models should be compared in other application cases. For example, in hopper flow and silo discharge. Can the proposed model predict the formation of the angle of repose on the surface of particles during silo discharge? Another interesting test case could be the free surface angle in a rotating drum.

This is a good comment. This article is invited by the editor and there is one month before the deadline. The times is enough to complete one or two new simulations. Unfortunately, because of China's strict pandemic prevention policy, we cannot come to university to complete the new simulation. In the future, we will consider completing these simulations. Of course, the effect of ICA must exist. Although the COR is the same, the lower Young's modulus means a longer contact time, and a faster energy dissipation rate. Hence, ICA is compared with SAA in this article, because both of them reach a stable state by accelerating dissipation.

4. No strong or weak scaling analyses are provided. The readers do not understand how the computational time changes versus the number of particles, and the computational cost of the model in parallel simulations on several computational cores.

It is difficult to give a quantitative answer. The acceleration effect is affected by many factors, even single threaded computing. Of course, there is no need to accelerate if the number of particles is not large. For parallel computing, the problem becomes more complex, it may related to computer hardware or communication between cores. Hence, some qualitative descriptions can be given. In the new manuscript, our answer is summarized in the end of section 6 (line 267-275)

My minor comments:

1.In equation 1, the dissipative force is generally presented as a function of relative velocity between the particles. Why is the dissipative force a function of normal overlap here?

This is a commonly used constant COR model. Many years ago, we also spent some time getting the source of this model. It is obtained from a harmonic vibration. Long story short, the viscous term is related to the elastic term, while the elastic term is related to the deformation, namely the overlap. The details can be found in Kawabara’s article (ref.[22])

2. Page 3, "To speed a DEM simulation using the collision model introduce ...", This assumption is only valid in gravity-driven flows. In other applications, it may cause non-negligible errors.

You are right. Now it is located in p5. Line 129, we polished the language and limited the application to accumulation problem due to gravitation. Similarly, we also modified the same error in line 20 in Abstract and line 284 in the Conclusion Section.

3. I do not understand Figure 3. The authors should elaborate more on explaining this figure. Why do we see two x axes for this figure? How does the number of particles correspond to the depth?

We added a sentence between line 165-167, Page 5 to explain this figure. This figure is obtained from simulation of glass bead with 1 cm. Thus, the depth can also be written as the number of particles.  

Reviewer 2 Report

Clearly and concisely written, the work is relevant to DEM and its applications. However, the authors must be very clear about the applicability and limitations of ICA+SAA. Make comparisons with monomodal, bimodal, multimodal, random, SCC, and FCC packing arrangements, as well as outputs from SAA algorithms.

Author Response

Response to Reviewer 2 Comments

 Point: Clearly and concisely written, the work is relevant to DEM and its applications. However, the authors must be very clear about the applicability and limitations of ICA+SAA. Make comparisons with monomodal, bimodal, multimodal, random, SCC, and FCC packing arrangements, as well as outputs from SAA algorithms.

Such kind comparison does not make too much sense. If our understanding is right, the monomodel and bimodel are generally suitable for multi-component particles. In this article, our particles are uniform in shape, although each particle is composed of several elements with different sizes. On the other hand, FCC and SCC is usually used for structures of crystals or metals. Generally spherical particles is used in these two models. While the particles in this article are complex in shape and in macroscopic scale.  If we misunderstood the points of these comments, we hope that some references can be list to help us know the points better. 

Reviewer 3 Report

In this manuscript, the authors describe a new algorithm introduced to generate the dense packing of spherical and non-spherical particles using the discrete element method.

 The comments related to the present manuscript are summarized below.

 1.  The literature review on the generation of dense initial packings by DEM is limited and should be enhanced. See, for example, the review in

Recarey C, Pérez I, Roselló R, Muniz M, Hernández E, Giraldo R, Oñate E. Advances in particle packing algorithms for generating the medium in the discrete element method. Computer Methods in Applied Mechanics and Engineering. 2019 Mar 1;345:336-62.

2.  Line 69: rij is defined as “a radius of each circular element composing an elliptical particle”. However, you did not mention until this line that you are dealing with elliptical particles. If the definition is correct, then you should introduce the particle shape before discussing the force model.

3.    Lines 90-93: Confirm the text. It seems that the formula is missing.

4.   Elaborate on how to define the thickness of the coating layer Dc for multi-sized particles and non-spherical particles.

5.   Conclusion, lines 245-246: Mention, if appropriate, the need to restore the value of Young’s module to the original one.

Author Response

Response to Reviewer 3 Comments

Dear reviewer,

We appreciate the comments from all reviews. We improved our manuscript according to these comments of each reviewer and marked them with different colors. Among these changes, those related to your comments are marked in blue. We hope that through these changes, the quality of the article can meet your requirements.

1. The literature review on the generation of dense initial packings by DEM is limited and should be enhanced. See, for example, the review in Recarey C, Pérez I, Roselló R, Muniz M, Hernández E, Giraldo R, Oñate E. Advances in particle packing algorithms for generating the medium in the discrete element method. Computer Methods in Applied Mechanics and Engineering. 2019 Mar 1;345:336-62.

Thanks for the information, we introduced this article near lines 23-28 page 1. And this article is also added to our reference, as ref.[4] in this new version.

2. Line 69: rij is defined as “a radius of each circular element composing an elliptical particle”. However, you did not mention until this line that you are dealing with elliptical particles. If the definition is correct, then you should introduce the particle shape before discussing the force model.

 Yes, we changed to order of these two sections. In addition, the introduction part in the end of sec.1 is also rewritten (line2 42-45 p.2)

3. Lines 90-93: Confirm the text. It seems that the formula is missing.

We checked the context. We do not think these is any formula missing.

4.  Elaborate on how to define the thickness of the coating layer Dc for multi-sized particles and non-spherical particles.

It is based on the maximum deformation of all particles, when they are densely packed, which is introduced in Sec.4. And the details is added near line 202-207, p.7 in this revised version.

5.  Conclusion, lines 245-246: Mention, if appropriate, the need to restore the value of Young’s module to the original one.

Thanks for the reviewer’s  reminding. It is our negligence. We added this part around lines 280-284 in the Conclusion Part.